# *Epichloë* Endophyte Alters Bacterial Nitrogen-Cycling Gene Abundance in the Rhizosphere Soil of Perennial Ryegrass

**DOI:** 10.3390/biology14070879

**Published:** 2025-07-18

**Authors:** Munire Maimaitiyiming, Yanxiang Huang, Letian Jia, Mofan Wu, Zhenjiang Chen

**Affiliations:** Key Laboratory of Grassland Livestock Industry Innovation, Ministry of Agriculture and Rural Aairs, State Key Laboratory of Grassland Agro-Ecosystems, Engineering Research Center of Grassland Industry, Ministry of Education, College of Pastoral Agriculture Science and Technology, Lanzhou University, Lanzhou 730020, China; munr2024@lzu.edu.cn (M.M.); 320220901551@lzu.edu.cn (Y.H.); 320220901561@lzu.edu.cn (L.J.); wumf2023@lzu.edu.cn (M.W.)

**Keywords:** *Epichloë endophyte*, *Lolium perenne*, sampling time, nitrogen-cycling gene, abundance and diversity

## Abstract

Perennial ryegrass (*Lolium perenne*) is a widespread forage and turf grass. *Epichloë* endophytes have been widely found in the above-ground tissues of grasses, and the effects of endophyte infection on ryegrass physiology and ecology have been relatively well studied. The response of the soil microbial community and nitrogen-cycling gene to this relationship has received much less attention. This study demonstrated the impacts of foliar fungal endophyte on the abundance and diversity of nitrogen-cycling genes involved in soil nitrification and denitrification. We found that the endophyte increased the concentrations of soil available nitrogen in ryegrass rhizosphere by promoting the abundance of the AOB-*amoA* gene involved in soil nitrification and decreasing the abundance of the *nosZ* gene involved in soil denitrification.

## 1. Introduction

Perennial ryegrass (*Lolium perenne* L.), widely cultivated in the cool season for forage supply and turfgrass, plays a crucial role in agricultural ecosystems [1]. It is known for its high nutritional value, cold tolerance and rapid regrowth after grazing or mowing. The symbiotic relationship between perennial ryegrass and the endophytic fungus *Epichloë festucae* var. *Lolii* has garnered significant attention due to its potential benefits for plant health and productivity.

Litter production and decomposition is a key ecosystem process that greatly influences the formation of soil organic matter, the release of nutrients for plants and microorganisms, CO_2_ and N_2_O fluxes and nutrient cycling in agricultural and forest ecosystems [2,3]. Nitrogen (N), P and approximately 60% of other mineral elements in forest ecosystems are typically recycled through litter decomposition [4,5,6,7]. An essential characteristic of the nitrogen cycle is that plant-available nitrogen predominantly derives from the microbial-mediated mineralization of organic matter (decomposition process) or biological nitrogen fixation [8,9]. The soil N cycle is mainly driven by a series of microbes (e.g., diazotrophs, nitrifiers and denitrifiers) which are collectively described as N-cycling microbes [10,11]. These nitrogen-cycling microorganisms mediate plant growth through biogeochemical transformations of nitrogen into bioavailable forms [nitrate (NO_3_^−^) or ammonium (NH_4_^+^)] that can be directly absorbed and used by plants [12]. Previous studies have demonstrated that litter addition (*Typha latifolia*) and litter quality significantly affected the abundance and diversity of two nitrifying genes (AOA-*amoA* and AOB-*amoA*) and three denitrifying genes (*nirS*, *nirK* and *nosZ*) involved in the N cycle [13,14].

*Epichloë* endophytes have been investigated with divergent outcomes regarding their impacts on host plant litter decomposition [15,16,17]. As with the effects of *Epichloë* endophytes on host–environmental stress factor interactions, effects on litter decomposition have been demonstrated to be less homogeneous, and have been found to vary from positive to negative [15,18]. Some toxic substances (e.g., endophyte alkaloids) produced by endophyte symbionts which have shown deterrent effects on herbivores result in litter becoming a low-quality substrate for many detritivores [19], and may exert a direct inhibitory effect on host litter decomposition. The changes in litter quality caused by *Epichloë* endophytes affect the decomposer community, and then affect litter decomposition [16]. Moreover, endophyte infection can have strong effects on the components of detrital food webs, including nematodes, earthworms, mites, collembola and soil microflora, and ultimately affect litter decomposition [16]. Results from both field and laboratory studies suggest that endophytes, which although mainly distributed in the above-ground parts of grasses, also affect the below-ground parts through influencing endophyte-mediated litter decomposition and root metabolism [16,20,21], may ultimately affect carbon and nutrient cycling. Previous studies have shown that leaf litter from *Epichloë* endophyte-infected ryegrass alters the abundance and diversity of the AOB-*amoA*, *nirK* and *nosZ* genes associated with soil N cycling in non-rhizosphere soil [22]. However, little is known about the effects of the *Epichloë* endophyte on the AOB-*amoA*, *nirK* and *nosZ* genes in the host rhizosphere soil of *L. perenne* plants after litter incorporation.

*Epichloë* endophytes (fungal genus *Epichloë*; formerly genus *Neotyphodium*) are a group of clavicipitaceous fungi that form hereditary symbioses with cool-season grasses from the subfamily Pooideae in temperate regions, and these symbioses are transmitted through successive host generations via vertical (seed) transmission [23,24]. *Epichloë* endophyte infection may increase host plants’ fitness by conferring stress tolerance to drought and poor soil, and/or resistance to herbivory and fungal diseases [25]. Fungal endophyte symbioses have generally been characterized as mutualistic [26,27]. However, it has been well established that leaf *Epichloë* endophytes can alter host phenotypes, expand ecological niches and increase fitness in some grasses, especially agronomic cultivars [28]. Documented effects of endophytes on wild host grasses in non-agricultural settings are highly variable [29,30]. The outcome of endophyte symbiosis appears to depend on the particular combination of host plant genotype, endophyte strain or species and environmental factors [31,32].

Here, we investigated the impact of the fungal endophyte *E. festucae* var. *Lolii* on the soil nitrogen-cycling gene involved in nitrification and denitrification. In particular, we aimed to understand how the soil nitrogen-cycling genes are influenced by the presence of endophytes, and whether these effects vary with different durations of litter incorporation in the rhizosphere soil. It was hypothesized that the effects of endophyte infection on soil nitrogen-cycling genes (AOB-*amoA*, *nirK* and *nosZ*) in rhizosphere soil would vary across litter incorporation times. The qPCR analysis was employed to estimate the total abundance of the AOB-*amoA*, *nirK* and *nosZ* genes coupled with amplicon sequencing to characterize the community composition of functional genes found in the rhizosphere soil collected from experimental endophyte-infected and endophyte-free *L. perenne* stands.

## 2. Materials and Methods

### 2.1. Plant Material

The ryegrass cultivar used was a turf-type grass selection of *L. perenne* Lanhei No. 1, which had been cultivated by our team over the last 18 years. This cultivar showed high tolerance to diseases [33] and low nutrient requirements [34]. Moreover, the infection frequency by the endophyte *E. festucae* var. *lolii* in seeds was 96.5% [35].

Seeds with high endophyte infection rate and seeds with genetically comparable and low endophyte infection rates were obtained by screening from primary material during three years in the field [35]. Briefly, seeds with an average endophyte infection rate of 62.5% were planted in September in 2014, and 128 single plants were harvested at maturity the next July. The 300 randomly selected seeds from each plant, along with all the tillers, were stained using aniline blue to determine the endophyte infection rate in seeds and tillers, respectively. Seeds with low (≤2%) and high (≥95%) infection rates in the tillers were designated as endophyte-infected (E+) and endophyte-free (E−) materials [35]. E+ and E− seed experimental plots were established at Yuzhong Experimental Station of Lanzhou University (104°12′ E, 35°85′ N, altitude 1400 m), Gansu Province, China, in 2015. In 2016 and 2017, the same method was used to determine the endophyte infection rate of each plant in E+ and E− seed fields, respectively. The high infection frequencies (E+) by *E. festucae* var. *lolii* in the tillers and seeds of *L.perenne* were 96.5%, and low infection frequencies (E−) were 1.1% [35]. In 2017, E+ and E− seeds of L. perenne were collected from the E+ and E− fields and stored at 4 °C to break seed dormancy and maintain endophyte viability.

### 2.2. Experiment Design

The experiments were conducted at the Yuzhong Experimental Station of Lanzhou University in Gansu Province, China, from 10 September 2017 to 1 November 2018. The test area was in a temperate continental climate, where the average annual temperature was 7.5 °C, the average annual rainfall was 358 mm and the frost-free period was 120 days/year during the test period [22]. The test site was evenly divided into 12 equivalent plots (3 m × 4 m). Initial soil was collected from experiment plots before planting by a five-point sampling method.

E+ and E− plots were established by bunch planting on pre-divided plots on 10 September 2017, using a 50 cm row spacing and distance between plants within rows. The experiment was laid out in a completely random design, with six replicates. Sprinkle irrigation was used, and weeds were removed by hand to avoid the effects of other organisms on soil properties. The endophyte infection status of seedlings in all plots was determined using the aniline blue staining method at 45 days after emergence, ensuring that the endophyte infection rates of plants were near 100% in E+ plots, and a mean 0% in E− plots. After 60 days, all plots were mowed to a height of 5 cm above the surface, and the harvested material was weighed to ensure that the weight of the returned fresh shoot litter was consistent across each plot. Leaf litters from E+ and E− (200 kg, each) were cut into 1 cm and incorporated by hand into soils at the respective row spacing of E+ and E− plots with a depth of 0–10 cm, respectively. Litter was incorporated average once every 120 days, totaling three times.

### 2.3. Soil Sampling and Chemical Analysis

The E+ and E− leaf litter materials were added to soil just after sampling. A total of four samplings were conducted: T_0_ (prior to first litter addition), T_1_ (post 120 d of 1st litter addition), T_2_ (post 120 d of 2nd litter addition) and T_3_ (post 120 d of 3rd litter addition). Four replicates of the endophyte and litter addition times were sampled for rhizosphere soil (i.e., soil adhering to plant roots) by five-point sampling. In each replicate, roots in multiple clumps of perennial ryegrass were moved out by inserting a soil corer in the center of the plant. Rhizosphere soil was collected by brushing roots with a sterilized paintbrush after gently shaking the soil around the roots [36]. Collected soil was immediately sieved (2 mm mesh) to remove any remaining root material and other impurities. Samples were divided into two parts; one part for DNA analysis was placed in a foam box with dry ice, transported to the laboratory and stored at −80 °C. The second part was brought back from the field at ambient temperature and stored at −20 °C to analyze soil chemical indicators.

Soil moisture content was measured gravimetrically after oven-drying soil (20 g) at 105 °C for 24 h [37]. Soil pH was determined using a pH meter (Sartorious PE10, Gottingen, Germany) with a suspension of soil and deionized water (1:2.5 = soil:water). Soil organic C (SOC 0.5 g) was determined through the K_2_CrO_7_-H_2_SO_4_ oxidation–reduction titration method [22]. To analyze soil ammonium nitrogen (NH_4_^+^-N) and nitrate nitrogen (NO_3_^−^-N), a fresh sample (5.0 g) was extracted using 20 mL of 2 M KCl, then shaken for 1 h at 200 rpm and filtered through Whatman No. 42 filter paper into a 50 cm plastic bottle [38]. The contents of NH_4_^+^-N and NO_3_^−^-N in the extracts were determined using a flow injection system (FIAstar 5000 Analyzer, Foss, Denmark). To analyze soil total N and total P, soil samples were digested with H_2_SO_4_ and a catalyst (CuSO_4_:K_2_SO_4_, 1:10 mixture) at 420 °C for 2 h on a digestion block (Hanon Instruments, Co., Ltd., Jinan, China), and a flow injection system (FIAstar 5000 Analyzer, Foss, Denmark) was used to determine the TN and TP contents. The ratios of C/N, C/P and N/P in the soil were calculated.

### 2.4. Soil DNA Extraction and Quantitative PCR

DNA was extracted from 500 mg freeze-dried (−80 °C) soil samples using the Fast-DNA^®^ Spin Kit of Soil Genome (MP Bio-medicals, Santa Ana, CA, USA), following the manufacturer’s instructions. The concentration and purity of extracted DNA was quantified using a Micro Nanodrop ND-1000 UV-Vis Spectrophotometer (NanoDrop™ 1000, Wilmington, DE, USA). To assess the change in total abundance of the AOB-*amoA, nirK* and *nosZ* genes, we used polymerase chain reaction (PCR)-based techniques to amplify following the approaches described in Chen et al. [22]. Briefly, we PCR-amplified DNA from three different gene markers to assess gene copy numbers of the AOB-*amoA, nirK* and *nosZ* genes. To examine functional genes involved in nitrification, the AOB-*amoA* gene was amplified and sequenced using b*amoA*1F (5′-GGGGTTTCTACTGGTGGT-3′) and b*amoA*2R (5′-CCCCTCKGSAAAGCCTTCTTC-3′) barcoded primers [39]. For functional genes involved in denitrification, the *nirK* and *nosZ* genes were amplified and sequenced using Cunir3F (5′-CGTCTAYCAYTCCGCVCC-3′) and Cunir3R (5′-GCCTCGATCAGRTTRTGG-3′) [40], *nosZ*-2F (5′-CGCRACGGCAASAAGGTSMSSGT-3′) and *nosZ*-2R (5′-CAKRTGCAKSGCRTGGCAGAA-3′) barcoded primers [41]. In all three cases, samples were amplified in quadruplicate. qPCR reactions from all samples were conducted on an CFX96 optical real-time detection system (CFX96TM Thermal Cycler, Foster City, CA, USA). High amplification efficiencies of 96.8% for the AOB-*amoA* gene, 95.2% for the *nirK* gene and 98.8% for the *nosZ* gene were obtained on account of standard curves, as calculated using the formula Eff = [10^(−1/slope)^ − 1] × 100%, respectively.

### 2.5. PCR Amplification and Sequencing

To characterize the diversity and composition of the AOB-*amoA, nirK* and *nosZ* genes in the rhizosphere soil of E+ and E− plants at T_1_ and T_3_ times for litter incorporation, we used PCR amplification and sequencing approaches. PCR products from all samples were recovered by gel cutting with the AxyPrepDNA Gel Recovery Kit (Axygen, Union City, CA, USA) following 2% agarose gel electrophoresis assay. Recovered PCR products were quantified using a QuantiFluor™ -ST blue fluorescent quantification system (Promega, Madison, WI, USA) and pooled together in equimolar concentrations. Amplicons were purified and concentrated using the QIAquick Gel Extraction Kit (Qiagen Sciences, Germantown, MD, USA). Samples were sequenced on an Illumina MiSeq PE300 instrument using a NEBNext^®^ Rapid DNA-Seq Kit (Illumina Inc., San Diego, CA, USA) at the Majorbio Bio-Pharm Technology Co., Ltd. (Shanghai, China), with separate runs for the AOB-*amoA, nirK* and *nosZ* genes amplicon pools.

### 2.6. Bioinformatics Analysis

Sequence analyses of the AOB-*amoA*, *nirK* and *nosZ* genes were performed on the Majorbio Cloud Platform (www.majorbio.com (accessed on 15 May 2024)) using the QIIME 2 pipeline (http://qiime.sourceforge.net/ (accessed on 15 May 2024)) [42]. Briefly, low-quality sequences were removed by fastp (length < 50 bp or with a quality value < 20 or having N bases) [43]. The extraction of non-repetitive sequences from optimized sequences (http://drive5.com/usearch/manual/dereplication.html (accessed on 23 April 2024)) and the removal of single sequences without duplicates (http://drive5.com/usearch/manual/singletons.html (accessed on 18 May 2024)) facilitated the reduction in redundant calculations in the analysis [44]. Filtered, high-quality PE reads were clustered for operational taxonomic unit (OTU) based on 97% similarity using the UPARSE v.11 pipeline (http://drive5.com/uparse/ (accessed on 20 May 2024)), and chimeras were removed in the clustering process to obtain representative sequences of OTU [41]. The OTU abundance tables of samples were generated by mapping all optimized sequences to OTU representative sequences and selecting sequences with a similarity of 97% or more to OTU representative sequences based on a pipeline of USEARCH v7.1 software. To obtain the classification information corresponding to each OTU of the AOB-*amoA*, *nirK* and *nosZ* genes in E+ and E− soils, RDP classifier and QIIME (Version 1.7.0) software with the functional gene database from GeneBank were used to perform a taxonomic analysis of the representative sequences of OTUs [45]. To compare the differences in AOB-*amoA*, *nirK* and *nosZ* genes in the rhizosphere soil of E+ and E+ plants in this study, OTUs at the phylum and genus levels were primarily utilized, as they are equivalent in taxonomic classification. The community diversity of the AOB-*amoA*, *nirK* and *nosZ* genes was presented using alpha (Shannon and chao1 indexes) diversity.

### 2.7. Statistical Analysis

The factor endophyte status (E) was utilized as a categorical variable with two levels: E+ (endophyte-infected plants) and E− (endophyte-free plants). Sampling times (T) were also categorical variables with four levels: T_0_, T_1_, T_2_, and T_3_. We used two-way ANOVA to analyze the effects of endophyte status (E) and sampling times (T) on soil pH, soil organic C (SOC), total nitrogen, total phosphorus, ammonium nitrogen (soil NH_4_^+^), nitrate nitrogen (soil NO_3_^−^), the C/N, C/P and N/P ratios and the absolute abundance and alpha diversity of the AOB-*amoA*, *nirK* and *nosZ* genes. The differences in rhizosphere soil between E+ and E− plants were assessed using Tukey’s b-test at *p* ≤ 0.05 when a significant effect was detected. All data analyses of variance (ANOVA) were performed in the software SPSS v. 20. The effects of endophyte status (E) on soil pH, SOC, total N, total P, NH_4_^+^, NO_3_^−^ and the C/N, C/P and N/P ratios were characterized under the same treatments (T_0_, T_1_, T_2_ and T_3_) through the independent sample *t*-test. We counted the species abundance of the AOB-*amoA*, *nirK* and *nosZ* genes in the rhizosphere soil of E+ and E− plants at the generic taxonomic level, and visualized the community composition by histogram. The Kruskal–Wallis H test was used to assess the significance level of differences in species abundance at *p* ≤ 0.05. Principal component analysis (PCA) was used to analyze the diversity of the AOB-*amoA*, *nirK* and *nosZ* genes at genus level between the rhizosphere soil of E+ and E− plants in two periods of sampling time (T_1_ and T_3_) to explore the similarities or differences in community composition between different grouped samples. To assess the correlation between microbial taxonomy and environmental factors, Pearson product-moment and Spearman’s rank-based correlation coefficients of the AOB-*amoA*, *nirK* and *nosZ* genes with environmental factors in the rhizosphere soil of E+ and E− plants was calculated using the Mantel test. We utilized canonical correspondence analysis (CCA) to characterize significant correlations between the top ten genera in the relative abundance of the AOB-*amoA*, *nirK* and *nosZ* genes and environmental factors, employing the CCA/RDA functions from the vegan package in R, as well as the Pearson product-moment correlation coefficient.

## 3. Results

### 3.1. Endophyte Infection Altered Soil Chemical Properties

Both endophyte status and sampling times exhibited significant effects on soil organic carbon (SOC), soil total P, soil NH_4_^+^, soil NO_3_^−^ and the C/N, C/P and N/P ratios, but only sampling times significantly affected soil pH and the concentrations of soil total N (Table 1 and Figure 1). Compared with the rhizosphere soil of E− plants, sampling time (T_2_) significantly increased soil total P, soil C/P and N/P ratios in the rhizosphere soil of E+ plants (Table 2 and Figure 1C,H,I), and sampling times (T_1_, T_2_ and T_3_) significantly increased the SOC and NH_4_^+^ concentrations in the rhizosphere soil of E+ plants (Table 2 and Figure 1D,E). The soil NO_3_^−^ content was significantly higher in the rhizosphere soil of E+ plants compared to that of E− plants at both T_2_ and T_3_ (Table 2 and Figure 1F), and the C/N ratio of E+ rhizosphere soil was significantly higher than that of E− rhizosphere soil at T_3_ (Table 2 and Figure 1H). Endophyte infection did not change soil pH and soil total N in comparison with endophyte non-infection under sampling times (T_1_, T_2_ and T_3_) (Table 2 and Figure 1A,B).

### 3.2. Endophyte Infection Affects Relative Abundances of Nitrification and Denitrification Functional Genes

There was a significant variation in the absolute abundance of the AOB-*amoA* gene between the rhizosphere soil of endophyte-infected (E+) plants and endophyte-free (E−) plants at both T_0_ and T_1_ times (Table 3). Endophyte infection significantly increased its abundance (Figure 2A). The combination of endophyte status and sampling times also significantly altered the relative abundance of the AOB-*amoA* gene (Table 4 and Figure 2D). The absolute and relative abundances of the *nirK* gene in the rhizosphere soil was only significantly affected by sampling times, rather than endophyte status or the interaction between both sampling times and endophyte status (Table 3 and Table 5 and Figure 2B,E). There was a significant effect of endophyte, sampling times and their interaction on the absolute abundance of the *nosZ* gene, which was significantly reduced at T_0_, T_1_ and T_3_ by endophyte infection (Table 4 and Figure 2C). The relative abundance of the *nosZ* gene in the rhizosphere soil was not significantly affected by endophyte status, sampling times and the interaction between both (Table 6 and Figure 2F).

### 3.3. Endophyte Effect on Alpha and Beta Diversity of the AOB-amoA, nirK and nosZ Functional Genes

Significant effects on the alpha diversity of the AOB-*amoA*, *nirK* and *nosZ* functional genes within communities were not observed based on endophyte status, sampling times or their interaction, as indicated by the Shannon and Chao1 indexes. (Table 3 and Figure 3). Endophyte infection significantly altered the beta diversity of the AOB-*amoA* functional gene in the T_1_ period of sampling times (Figure 4A), but the beta diversity of the *nirK* and *nosZ* functional genes did not show any significant difference between the rhizosphere soil of E+ and E− plants at different sampling times (T_1_ and T_3_) (Figure 3B,C).

### 3.4. Relationships Between the AOB-amoA, nirK and nosZ Functional Genes with Soil Properties

Major gradients in the abundance and diversity of AOB-*amoA* gene differentiation were visualized by CCA, with the first two axes explaining 95.75% of the variance in the abundance and diversity of the AOB-*amoA* gene following environmental properties (Figure 5A). CCA confirmed the larger effect of soil pH (*p* = 0.008) and soil NH_4_^+^ (*p* = 0.03) on the abundance and diversity of the AOB-*amoA* gene compared to the other environmental parameters, and the contribution of the two amounted to 34.3% and 30.8%, respectively (Figure 5A). Analyzing the abundance and diversity of the *nirK* gene indicated that changes in soil properties accounted for approximately 70.38% of the variation along the first axis and 16.69% along the second axis, with soil NH_4_^+^ (*p* = 0.002, 50.4%) and soil C/P ratio (*p* = 0.012, 15.8%) having a strong effect (Figure 5B). Our CCA analysis of the *nosZ* functional gene abundance indicated that changes in soil properties, particularly with litter returning, explained approximately 85.51% of the variation along the first axis and 11.83% along the second axis (Figure 5C). There was a clear differentiation of the abundance and diversity of the *nosZ* gene between endophyte-infected (E+) and endophyte-free (E−) plants at different litter addition times (T_1_ and T_3_) along with the first principal component, indicated by an obvious clustering of the study plots (Figure 5C). For the *nosZ* gene, soil pH (*p* = 0.03, 20.6%) and soil N/P ratio (*p* = 0.002, 68.5%) had significant impacts on the abundance and diversity compared to the other environmental parameters (Figure 5C).

## 4. Discussion

Shifts in soil microbial communities caused by plant symbiotic microorganisms, especially foliar *Epichloë* endophytes, have a direct or indirect impact on ecosystem functioning and soil fertility and crop productivity, which makes it a necessity to study how soil microbiomes respond to symbiotic microorganisms. Numerous studies have been conducted to understand the effect of foliar fungal endophytes on soil microbial communities, as well as on their interactions [46,47,48]. These studies provide essential knowledge of how *Epichloë* endophytes affect soil bacterial and fungal communities [49,50,51,52]. The current study revealed the effects of the *Epichloë* endophyte on the abundance and diversity of soil nitrogen-cycling genes (e.g., the AOB-*amoA*, *nirK* and *nosZ* functional genes) in host rhizosphere soil. The results indicated that infection with the endophyte *Epichloë festucae* var. *lolii* significantly increased the absolute abundance of the AOB-*amoA* gene and significantly decreased the absolute abundance of the *nosZ* gene in the rhizosphere soil of the host plant *L. perenne* at various sampling times (T_0_, T_1_ and T_3_), but not significantly affected the absolute abundance of the *nirK* gene. Meanwhile, the relative abundance and beta diversity of the AOB-*amoA* gene and the relative abundance of the *nosZ* gene in the host rhizosphere soil were affected by endophyte infection. These results were similar to our previous experiment, in which the leaf litter perennial ryegrass containing the fungal endophyte significantly altered the abundance and diversity of the AOB-*amoA*, *nirK* and *nosZ* genes in non-rhizosphere soil [22]. Concurrently, the absolute abundance of the AOB-*amoA* gene in host rhizosphere soil at the T_1_ sampling time, and the absolute abundance of the *nosZ* gene at T_3_ and T_1_ sampling times, were higher than those at T_0_ time. Overall, the *Epichloë* endophyte had an effect on the absolute and relative abundances and beta diversity of the AOB-*amoA* gene and the relative abundance of the *nosZ* gene in the rhizosphere soil of *L. perenne* plants.

Several reported studies showed that the *Epichloë* endophyte in *Lolium multiflorum* caused changes in the soil bacterial community structure by modifying host rhizodeposition [46]. Infection by *Epichloë* endophytes can modify litter decomposition by altering both the quantity and quality of the litter, or by changing the decomposer communities [14,17]. Our previous study also demonstrated that litter containing the fungal endophyte *E. festucae* var. *lolii* significantly altered the abundance and diversity of the AOB-*amoA* functional genes in non-rhizosphere soil compared to endophyte-free litter [22]. We found significant differences in the absolute and relative abundances and beta diversity of the AOB-*amoA* gene at T_1_ time between the rhizosphere soil of endophyte-infected plants and endophyte-free plants. Consistent with the previous findings by Jin et al. [53], the absolute abundance of the AOB-*amoA* gene was increased in the rhizosphere soil of *Achnatherum inebrians* by endophyte infection. These findings suggested that which pathways are mediated by endophytes impacts the abundance and beta diversity of the AOB-*amoA* gene involved in soil nitrification, which in turn affects nitrogen transformation. Extensive studies have confirmed that soil nitrification is strongly correlated with environmental factors, including aeration conditions, texture, temperature, soil moisture, pH value and fertilization factors. Soil pH is the main factor affecting the AOA-*amoA* or AOB-*amoA* gene communities [54]. Our study supports the previous literature suggesting that the abundance and diversity of the AOB-*amoA* gene was significantly correlated with soil pH and NH_4_^+^.

Although the denitrifying genes (*nirS*, *nirK* and *nosZ*) are the most extensively studied as markers for the composition and population of soil microbial communities in forest, wetland, pasture and agricultural soil [55,56], they are rare in the rhizosphere soil of endophyte-infected plants, raising concerns regarding *Epichloë* endophytes in beneficial effects of denitrification in soil. Our previous findings demonstrated that litter containing the fungal endophyte significantly reduced the abundance and alpha diversity of the *nirK* and *nosZ* genes involved in soil denitrification [22]. Jin et al. [53] showed that the change in the abundance of denitrification genes *nirK* and *nosZ* by *Epichloë* endophyte was dependent on host ecotypes. In the current study, no significant alterations in the abundance (Figure 3B,E) and diversity (Figure 4B and Figure 5B) of the *nirK* gene were observed between the rhizosphere soil of endophyte-infected plants and that of endophyte-free plants across various sampling times. Because of its importance as the final step in denitrification, *nosZ* was chosen to determine the effect of the *Epichloë* endophyte. The abundance but not diversity significantly changed in rhizosphere soil of the host ryegrass under *Epichloë* endophyte infection, compared to the rhizosphere soil of endophyte-free plants under different sampling times (T_0_, T_1_ and T_3_). A possible explanation for this finding could be that the effect of the *Epichloë* endophyte on *nirK nosZ* genes might related to host types, experimental treatments and soil types (e.g., rhizosphere soils and bulk soils).

This study provides new insights into how plant–endophyte interactions regulate soil microbial functional groups to influence nitrogen cycling. Future research should integrate multi-omics approaches (e.g., metagenomics, metabolomics) to elucidate the underlying molecular mechanisms and validate these findings across diverse soil environments and host systems.

## 5. Conclusions

To our knowledge, the response of the nitrogen-cycling gene in host rhizosphere soil to a symbiotic relationship between *Lolium perenne* plants and the fungal endophyte *Epichloë festucae* var. *lolii.* has received little attention. This study demonstrated the impacts of foliar fungal endophyte on the abundance and diversity of nitrogen-cycling genes involved in soil nitrification and denitrification. We found that the endophyte increased the concentrations of soil available nitrogen in ryegrass rhizosphere by promoting the abundance of the AOB-*amoA* gene involved in soil nitrification and decreasing the abundance of the *nosZ* gene involved in soil denitrification. The changes in the abundance and diversity of the AOB-*amoA* gene were associated with soil pH and NH_4_^+^ concentration, and the abundance and diversity of the *nosZ* gene was associated with soil pH and soil N/P ratio. The findings from this study may contribute to a new understanding of how endophytes affect functional genes involved in the soil nitrogen cycle, as well as the associated mechanisms in the rhizosphere soil of host plants. Further studies are necessary to elucidate the correlation among nitrogen-cycling gene, soil N transformation, and *Epichloë* endophyte-mediated litter decomposition and root exudation.

## Figures and Tables

**Figure 1 biology-14-00879-f001:**
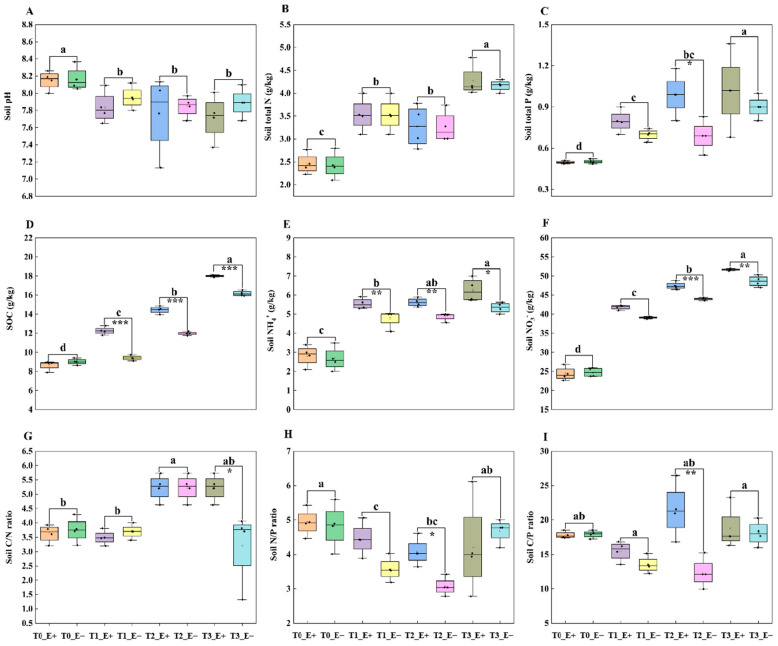
Effects of *Epichloë* endophyte and sampling times (T_0_, T_1_, T_2_ and T_3_) on rhizosphere soil properties of *Lolium perenne*. (**A**): Soil pH, (**B**): soil total N, (**C**): soil total P, (**D**): SOC, (**E**): soil NH_4_^+^, (**F**): soil NO_3_^−^, (**G**): the C/N ratio, (**H**): the N/P ratio and (**I**): the N/P ratio. Values are mean ± standard error (SE). Different lowercase letters indicate significant differences at *p* ≤ 0.05, among different sampling times (T0, T1, T2 and T3). Also, *, ** and *** mean significant difference at *p* ≤ 0.05%, *p* ≤ 0.01% and *p* ≤ 0.001%, respectively (independent *t*-test), between endophyte-infected (E+) and endophyte-free (E−) plant rhizosphere soil under the same addition times. Same as below.

**Figure 2 biology-14-00879-f002:**
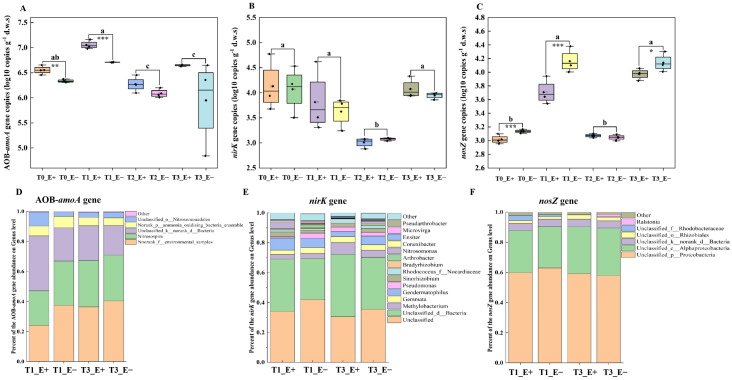
Effects of *Epichloë* endophyte on the absolute and relative abundances (at the genus level) of the AOB-*amoA*, *nirK* and *nosZ* genes in rhizosphere soil properties of *Lolium perenne* under sampling times (T_1_ and T_3_). (**A**,**D**): The AOB-*amoA* gene, (**B**,**E**): the *nirK* gene and (**C**,**F**): *nosZ* gene.Also, Different lowercase letters indicate significant differences at *p* ≤ 0.05, among different sampling times (T0, T1, T2 and T3). *, ** and *** mean significant difference at *p* ≤ 0.05%, *p* ≤ 0.01% and *p* ≤ 0.001%.

**Figure 3 biology-14-00879-f003:**
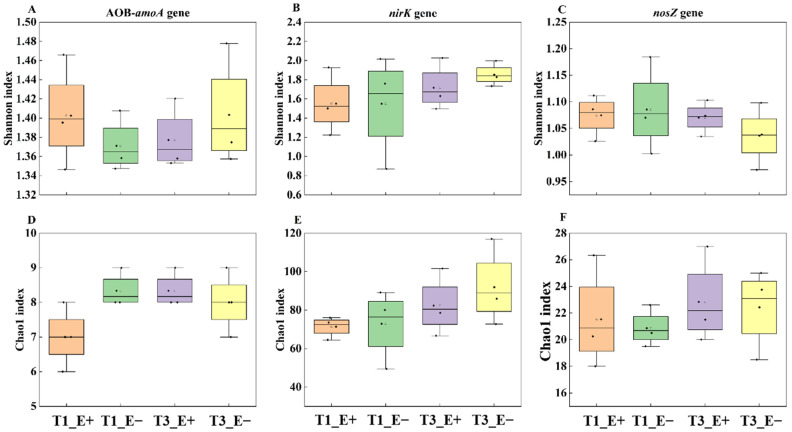
Effects of two levels of sampling times (T_1_ and T_3_) on Shannon and Chao1 indexes (at the genus level) of the AOB-*amoA*, *nirK* and *nosZ* genes in the E+ and E− rhizosphere soils under litter addition (T_1_ and T_3_). (**A**,**D**): The AOB-*amoA* gene, (**B**,**E**): the *nirK* gene and (**C**,**F**): *nosZ* gene.

**Figure 4 biology-14-00879-f004:**
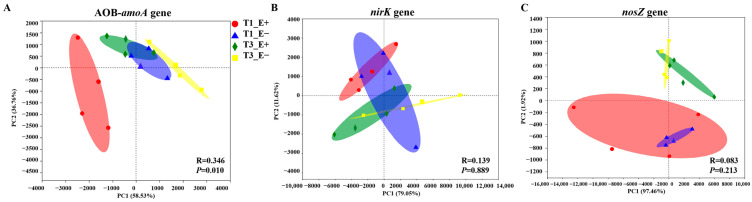
Principal component analysis (PCA) (at the genus level) of the AOB-*amoA*, *nirK* and *nosZ* genes in the E+ and E− rhizosphere soils under sampling times (T_1_ and T_3_). (**A**): The AOB-*amoA* gene, (**B**): the *nirK* gene and (**C**): *nosZ* gene.

**Figure 5 biology-14-00879-f005:**
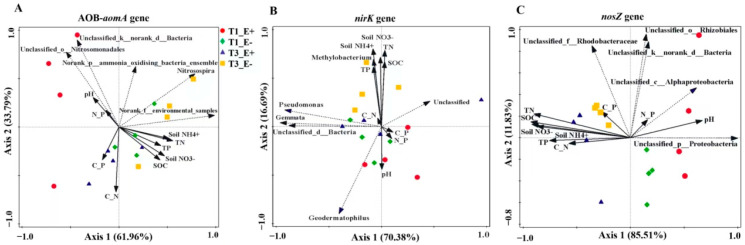
Canonical correlation analysis (at the genus level) of relative abundance and diversity of the AOB-*amoA*, *nirK* and *nosZ* genes in the E+ and E− rhizosphere soils under sampling times. Environmental factors include soil pH, SOC (soil organic carbon), TN (soil total nitrogen), TP (soil total phosphorus), soil NH_4_^+^, soil NO_3_^−^, C_N (the SOC and TN ratio) and C_P (the SOC and TP ratio). (**A**): The AOB-*amoA* gene, (**B**): the *nirK* gene and (**C**): *nosZ* gene.

**Table 1 biology-14-00879-t001:** Two-way analysis of variance (two-way ANOVA) for the effect of endophyte status (E) and sampling times (T) on soil pH, soil organic carbon (SOC), total nitrogen, total phosphorus, ammonium nitrogen (soil NH_4_^+^), nitrate nitrogen (soil NO_3_^−^) and the C/N, C/P and N/P ratios in rhizosphere soil of *Lolium perenne*.

Treatment	df	Soil pH		SOC		Soil Total N		Soil Total P		Soil NH_4_^+^		Soil NO_3_^−^		C/N Ratio		C/P Ratio		N/P Ratio	
		F	*p*	F	*p*	F	*p*	F	*p*	F	*p*	F	*p*	F	*p*	F	*p*	F	*p*
Endophyte (E)	1	1.467	0.238	384.473	<0.000	0.431	0.518	8.130	0.009	17.843	<0.000	38.522	<0.000	7.265	0.013	14.471	0.001	2.853	0.048
Litter addition times (T)	3	4.483	0.012	1536.522	<0.000	5.195	0.007	18.725	<0.000	78.311	<0.000	979.759	<0.000	5.564	0.005	5.125	0.007	6.337	0.003
E x T	3	0.170	0.916	55.246	<0.000	1.692	0.195	1.981	0.144	1.150	0.349	6.158	0.003	4.785	0.009	7.675	0.001	2.396	0.093

**Table 2 biology-14-00879-t002:** The independent sample *t*-test for the effects of endophyte status (E) on soil pH, SOC, total N, total P, NH_4_^+^, NO_3_^−^ and the C/N, C/P and N/P ratios in rhizosphere soil of *Lolium perenne* under different sampling times (T_0_, T_1_, T_2_ and T_3_).

Times		Soil pH	SOC	Soil Total N	Soil Total P	Soil NH_4_^+^	Soil NO_3_^−^	Soil C/N Ratio	Soil C/P Ratio	Soil N/P Ratio
T_0_	df	7	7	7	7	6	6	7	6	6
	F	0.716	5.383	1.259	4.645	0.037	0.110	0.000	0.001	0.000
	*p*	0.453	0.508	0.426	0.435	0.701	0.718	0.876	0.758	0.571
T_1_	df	5	5	5	5	6	6	6	6	6
	F	0.029	0.059	1.036	0.175	0.885	8.394	0.119	0.148	8.568
	*p*	0.438	<0.000	0.056	0.131	0.001	0.234	0.366	0.080	0.402
T_2_	df	6	6	6	6	6	6	6	6	6
	F	2.934	0.955	0.021	0.135	0.043	1.931	2.076	0.552	0.363
	*p*	0.743	<0.000	0.171	0.021	0.003	<0.000	0.264	0.007	0.006
T_3_	df	6	6	6	6	6	6	6	6	6
	F	0.427	3.388	8.273	1.376	5.960	8.294	1.657	1.148	2.596
	*p*	0.314	<0.000	0.326	0.439	0.049	0.023	0.017	0.733	0.536

**Table 3 biology-14-00879-t003:** Two-way ANOVA for the effect of endophyte status (E) and sampling times (T) on alpha diversity indexes (Shannon and Chao1) of the AOB-amoA, nirK and nosZ genes at the genus level.

Treatment	df	AOB-*amoA* Gene				*nirK* Gene				*nosZ* Gene			
		Shannon		Chao1		Shannon		Chao1		Shannon		Chao1	
		F	*p*	F	*p*	F	*p*	F	*p*	F	*p*	F	*p*
Endophyte (E)	1	0.008	0.930	1.125	0.320	1.142	0.316	2.179	0.178	0.552	0.479	0.525	0.489
Litter addition times (T)	1	0.013	0.911	1.125	0.320	0.092	0.769	0.002	0.969	0.103	0.756	0.075	0.791
E x T	1	0.969	0.354	3.125	0.115	0.096	0.764	0.065	0.805	0.397	0.546	0.004	0.952

**Table 4 biology-14-00879-t004:** Two-way ANOVA for the effect of endophyte status (E) and sampling times (T) on the relative abundance of the AOB-amoA gene at the genus level.

Treatment	df	Nnorank_f__Environmental_Samples		*Nitrosospira*		Unclassified_k__Norank_d__Bacteria		Norank_p__Ammonia_Oxidising_Bacteria_Ensemble		Unclassified_o__Nitrosomonadales		Other	
		F	*p*	F	*p*	F	*p*	F	*p*	F	*p*	F	*p*
Endophyte (E)	1	10.253	0.013	4.979	0.056	62.421	<0.000	0.068	0.801	7.263	0.027	0.685	0.432
Litter addition times (T)	1	7.834	0.023	11.030	0.011	49.618	<0.000	3.745	0.089	5.196	0.052	0.992	0.348
E x T	1	2.990	0.122	6.383	0.035	24.413	0.001	0.698	0.428	8.323	0.020	1.013	0.344

**Table 5 biology-14-00879-t005:** Two-way ANOVA for the effect of endophyte status (E) and sampling times (T) on the relative abundance of the *nirK* gene at the genus level.

Treatment	df	Unclassified		Unclassified_d__Bacteria		Methylobacterium		Gemmata		Geodermatophilus		Pseudomonas		Sinorhizobium		Rhodococcus_f__Nocardiaceae	
		F	*p*	F	*p*	F	*p*	F	*p*	F	*p*	F	*p*	F	*p*	F	*p*
Endophyte (E)	1	0.347	0.572	3.198	0.112	0.405	0.542	0.127	0.730	1.112	0.322	0.457	0.518	0.956	0.357	1.362	0.277
Litter addition times (T)	1	0.117	0.742	0.217	0.654	8.564	0.019	0.165	0.695	5.866	0.042	0.165	0.695	0.139	0.719	1.104	0.324
E x T	1	0.002	0.963	1.206	0.304	0.835	0.388	0.003	0.955	0.006	0.942	0.795	0.399	0.012	0.915	0.827	0.390
**Treatment**	**df**	**Bradyrhizobium**		**Arthrobacter**		**Nitrosomonas**		**Conexibacter**		**Ensiter**		**Microvirga**		**Pseudarthrobacter**		**Other**	
		**F**	** *p* **	**F**	** *p* **	**F**	** *p* **	**F**	** *p* **	**F**	** *p* **	**F**	** *p* **	**F**	** *p* **	**F**	** *p* **
Endophyte (E)	1	1.062	0.333	0.649	0.444	1.067	0.332	0.724	0.420	1.692	0.229	0.245	0.634	0.870	0.378	0.107	0.752
Litter addition times (T)	1	0.724	0.420	1.774	0.220	1.174	0.310	0.188	0.676	1.371	0.275	0.557	0.477	1.476	0.259	0.094	0.767
E x T	1	1.020	0.342	1.018	0.343	1.342	0.280	0.839	0.386	0.000	0.987	0.154	0.705	0.870	0.378	0.222	0.650

**Table 6 biology-14-00879-t006:** Two-way ANOVA for the effect of endophyte status (E) and sampling times (T) on the relative abundance of the *nosZ* gene at the genus level.

Treatment	df	Unclassified_p__Proteobacteria		Unclassified_c__Alphaproteobacteria		Unclassified_k__Norank_d__Bacteria		Unclassified_o__Rhizobiales		Unclassified_f__Rhodobacteraceae		Ralstonia		Other	
		F	*p*	F	*p*	F	*p*	F	*p*	F	*p*	F	*p*	F	*p*
Endophyte (E)	1	1.308	0.286	0.009	0.926	0.000	0.993	9.643	0.015	8.860	0.018	1.790	0.218	1.001	0.346
Litter addition times (T)	1	12.022	0.008	13.770	0.006	0.086	0.776	43.165	<0.000	35.847	<0.000	1.790	0.218	3.771	0.088
E x T	1	7.126	0.028	0.279	0.612	0.482	0.507	1.455	0.262	8.976	0.017	1.790	0.218	0.756	0.410

## Data Availability

The nucleotide sequences of the AOB-*amoA*, *nirK* and *nosZ* functional genes from this study have been deposited the NCBI sequence read archive (Bioproject: PRJNA861957, PRJNA861959 and PRJNA861960, https://www.ncbi.nlm.nih.gov/bioproject/PRJNA861957 (accessed on 25 July 2022), PRJNA861959 and PRJNA861960, respectively).

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
