# Peer review of "Epichloë Endophyte Alters Bacterial Nitrogen-Cycling Gene Abundance in the Rhizosphere Soil of Perennial Ryegrass"

_biology, 2025, doi:10.3390/biology14070879_

Round 1
Reviewer 1 Report
Comments and Suggestions for Authors
The ms entitled “Epichloë endophyte alters bacterial nitrogen‐cycling gene abundance in the rhizosphere soil of perennial ryegrass” emphasized on abunance and diversity variation of the AOB-amoA, nirK and nosZ functional genes in the rhizosphere soil of the endophyte-ryegrass symbiosis following litter addition. The topic is interested but the authors have to answer on the following questions and improve their ms before it can go for next step.
Why the authors did not repeat their experiment twice at least to show if the data were changed or similar or ..?
I strongly recommend authors to provide some pictures for their work, this can make it great.
L34: All abbreviations there or somewhere else have to be defined in the first mention, even if they are known. See P, SOC, NH4 + and NO3
L36: P < 0.05, you can write it like that, but it must be P ≤ 0.05. Please revise this issue in the whole ms.
In abstract: please add some important values of your key results.
L54: N, P and 60% of the other mineral elements required for plant growth come from litter [3]. Are you sure? This is very old reference 1989! Please check and update.
L67-69 Please reduce the number of references used there, why 5 references? Two are enough.
Introduction section: I am wondering that the introduction does NOT have enough information and references on ryegrass, although it is one of the main keys for your current study. Please revise this issue and add some text about ryegrass.
L123-124 The place where the experiment was done should be moved to earlier place in material and methods. Yuzhong Experimental Station of Lanzhou University (104°12′ E, 35°85′ N, altitude 1400m), Gansu Province, China
In material and methods do not repeat yourself in writing, I saw repeated text in some places there. Please remove and update.
Results section:
The authors have to add the units of different measurements that were presented in Tables or Figures where are available.
The authors should make it clear and Tables are not well presented. The authors should add values with at least two decimal. Statistical analysis are not well done, please check because I found some mistakes in Tables 1-3.
In discussion section, I wish the authors to focus on the mechanism of their treatments on what they have found, this is an important issue. Now, the discussion is superficial written.
References: Authors have to add some recent references.
-
Reviewer 2 Report
Comments and Suggestions for Authors
This manuscript explores how Epichloë festucae var. lolii, a fungal endophyte commonly found in perennial ryegrass (Lolium perenne), affects the abundance and diversity of soil nitrogen-cycling genes (AOB-amoA, nirK, and nosZ) in the plant rhizosphere across multiple litter incorporation time points. The study shows that endophyte presence promotes nitrification (increased AOB-amoA abundance) and suppresses denitrification (decreased nosZ abundance). The work provides valuable insights into plant-microbe-soil interactions with implications for sustainable nitrogen management in grassland ecosystems.
1. Could the authors explain whether the observed effects are due to changes in root exudation or microbial interactions caused by endophyte-modified litter?
2. Could the authors clarify whether seasonal variation may have influenced results across T0–T3 sampling times?
3. Can the authors clarify or reformulate the main hypothesis in a more testable, predictive format?
For example: “We hypothesised that endophyte infection would increase AOB-amoA gene abundance due to enhanced soil NH₄⁺ availability.”
4. Why was a bulk soil (non-rhizosphere) control not included for comparison? Including such a control could help separate root-mediated vs. litter-mediated microbial responses.
5. Could the authors explain why sampling was done at 120-day intervals? Was this based on previous studies or pilot data?
6. Can the authors elaborate on the mechanistic link between endophyte-mediated litter changes and rhizosphere microbial shifts? Are these changes due to direct chemical inputs from litter or altered root exudates?
7. Could seasonal or environmental factors confound the time-point comparisons? Is temperature, rainfall, or soil moisture variability accounted for across T0–T3?
8. What are the next steps? Would metagenomic or transcriptomic approaches better capture functional dynamics beyond gene abundance?
Author Response
Comments1:Could the authors explain whether the observed effects are due to changes in root exudation or microbial interactions caused by endophyte-modified litter?
Response1:Thank you for raising this insightful question regarding the mechanistic basis of the observed effects.In response,we will explain this result in the discussion of the content of the second part whether these effects are due to changes in root exudation or microbial interactions caused by endophyte-modified litter .Our previous study also demonstrated that litter containing the fungal endophyte E. festucae var. lolii significantly altered the abundance and diversity of the AOB-amoA functional genes in non-rhizosphere soil compared to endophyte-free litter. We found significant differences in the absolute and relative abundances, and beta diversity of the AOB-amoA gene at T1 time between rhizosphere soil of endophyte-infected plants and endophyte-free plants. Consistent with the previous findings by Jin et al., the absolute abundance of the AOB-amoA gene was increased in rhizosphere soil of Achnatherum inebrians by endophyte infection. These findings suggested that what pathways mediated by endophyte impact abundance and beta diversity of AOB-amoA gene involved in soil nitrification, which in turn affects nitrogen transformation.
Comments2:Could the authors clarify whether seasonal variation may have influenced results across T0–T3 sampling times?
Response2:For the experiment , litter addition and all other treatments were applied uniformly across sampling times (T0–T3); Identical sampling protocols were maintained throughout the study period. The purpose of this study is it was aimed to know how soil nitrogen‐cycling gene are affected by endophyte presence, and whether the effects differ across different litter incorporation times in rhizosphere soil.
Comments3:Can the authors clarify or reformulate the main hypothesis in a more testable, predictive format?For example: “We hypothesised that endophyte infection would increase AOB-amoA gene abundance due to enhanced soil NH₄⁺ availability.”
Response3: Thank you for your valuable suggestions regarding the formulation of our research hypothesis. We have carefully considered your comments on enhancing the testability of the hypothesis and engaged in thorough team discussions. After deliberation, we have decided to retain the original hypothesis structure based on the following considerations:The primary goal of this study is to exploratorily unravel the tripartite interactions among between the foliar endophyte E. festucae var. Lolii, L. perenne and soil nitrogen‐cycling gene (rather than validating a single causal pathway). The open-ended framework of our original hypothesis better aligns with this exploratory research design.
Comments4:Why was a bulk soil (non-rhizosphere) control not included for comparison? Including such a control could help separate root-mediated vs. litter-mediated microbial responses.
Response4:We appreciate the opportunity to clarify our experimental design regarding the inclusion of bulk soil controls. This study specifically targets rhizosphere-mediated processes, and while we fully acknowledge the scientific value of bulk soil controls, such non-rhizosphere control groups have been addressed in prior published work.
Comments5:Could the authors explain why sampling was done at 120-day intervals? Was this based on previous studies or pilot data?
Response5:Conducting experiments every three months during the four growing seasons can better avoid soil disturbance caused by frequent sampling and also ensure the time required for the succession of microbial communities.
Comments6:Can the authors elaborate on the mechanistic link between endophyte-mediated litter changes and rhizosphere microbial shifts? Are these changes due to direct chemical inputs from litter or altered root exudates?
Response6:Epichloë endophytes infection significantly increased the absolute abundance of the AOB-amoA gene in the rhizosphere soil of host perennial ryegrass across different sampling periods. Concurrently, it significantly reduced the absolute abundance of the nosZ gene , while no significant effect was observed on the nirK gene. Notably, the absolute abundance of the AOB-amoA gene in host rhizosphere soil at T1 sampling time, and the absolute abundance of nosZ gene at T3 and T1 sampling times were higher than T0 time.
Comments7:Could seasonal or environmental factors confound the time-point comparisons? Is temperature, rainfall, or soil moisture variability accounted for across T0–T3?
Response7:Environmental or seasonal factors will not be confused. In this experiment, we did not take into account the soil moisture changes from T0 to T3. We will pay attention to this point in future experiments.
Comments8:What are the next steps? Would metagenomic or transcriptomic approaches better capture functional dynamics beyond gene abundance?
Response8:The next phase of this study will investigate the functional shifts in rhizosphere soil within the endophytic bacteria-ryegrass symbiotic system following litter amendment. Specifically, we will compare soil functional profiles before and after litter addition, with a focus on nitrogen-cycling dynamics. Further research is essential to clarify the interplay between nitrogen-cycling genes, soil nitrogen transformation processes, and Epichloë endophyte-mediated litter decomposition coupled with root exudation.
Round 2
Reviewer 1 Report
Comments and Suggestions for Authors
The ms has been improved but still the discussion can be improved, I will leave this issue for the editor.
Author Response
Comments1:The ms has been improved but still the discussion can be improved, I will leave this issue for the editor.
Response1:
we have added some sentences (see L448-452) with the modifications highlighted in red in the revised manuscript.
This study provides new insights into how plant-endophyte interactions regulate soil microbial functional groups to influence nitrogen cycling. Future research should integrate multi-omics approaches (e.g., metagenomics, metabolomics) to elucidate the underlying molecular mechanisms and validate these findings across diverse soil environments and host systems.